# Fluorescence Cross-Correlation Spectroscopy Yields True Affinity and Binding Kinetics of *Plasmodium* Lactate Transport Inhibitors

**DOI:** 10.3390/ph14080757

**Published:** 2021-08-02

**Authors:** Iga Jakobowska, Frank Becker, Stefano Minguzzi, Kerrin Hansen, Björn Henke, Nathan Hugo Epalle, Eric Beitz, Stefan Hannus

**Affiliations:** 1Intana Bioscience GmbH, Lochhamer Str. 29a, 82152 Planegg, Germany; iga.jakobowska@intana.de (I.J.); frank.becker@intana.de (F.B.); stefano.minguzzi@intana.de (S.M.); kerrin.hansen@intana.de (K.H.); 2Pharmaceutical Institute, Christian-Albrechts-University of Kiel, Gutenbergstr. 76, 24118 Kiel, Germany; bhenke@pharmazie.uni-kiel.de (B.H.); nepalle@pharmazie.uni-kiel.de (N.H.E.)

**Keywords:** malaria, lactate, transport, inhibitor, fluorescence cross-correlation spectroscopy, binding affinity, formate-nitrite transporter

## Abstract

Blocking lactate export in the parasitic protozoan *Plasmodium falciparum* is a novel strategy to combat malaria. We discovered small drug-like molecules that inhibit the sole plasmodial lactate transporter, PfFNT, and kill parasites in culture. The pentafluoro-3-hydroxy-pent-2-en-1-one BH296 blocks PfFNT with nanomolar efficiency but an in vitro selected PfFNT G107S mutation confers resistance against the drug. We circumvented the mutation by introducing a nitrogen atom as a hydrogen bond acceptor site into the aromatic ring of the inhibitor yielding BH267.meta. The current PfFNT inhibitor efficiency values were derived from yeast-based lactate transport assays, yet direct affinity and binding kinetics data are missing. Here, we expressed PfFNT fused with a green fluorescent protein in human embryonic kidney cells and generated fluorescent derivatives of the inhibitors, BH296 and BH267.meta. Using confocal imaging, we confirmed the location of the proposed binding site at the cytosolic transporter entry site. We then carried out fluorescence cross-correlation spectroscopy measurements to assign true K_i_-values, as well as k_on_ and k_off_ rate constants for inhibitor binding to PfFNT wildtype and the G107S mutant. BH296 and BH267.meta gave similar rate constants for binding to PfFNT wildtype. BH296 was inactive on PfFNT G107S, whereas BH267.meta bound the mutant protein albeit with weaker affinity than to PfFNT wildtype. Eventually, using a set of PfFNT inhibitor compounds, we found a robust correlation of the results from the biophysical FCCS binding assay to inhibition data of the functional transport assay.

## 1. Introduction

Asexual plasmodia draw vital energy from glycolysis [1]. The pathway relies on the swift release of the metabolic end product lactate and its accompanying proton. For this purpose, in the blood stage, the parasites express a single lactate/H^+^ co-transporter, PfFNT, of the exclusively microbial formate-nitrite transporter family [2]. Inhibition of PfFNT by drug-like small molecules from the MMV Malaria Box [3] (hit compounds MMV007839 and MMV000972) led to the cessation of the energy metabolism and acidification of the cytosol, killing the parasite [4,5]. Cytosolic pH disturbance by PfFNT inhibitors supposedly has additional multitarget effects on the functionality of enzymes and transporters, constituting a known novel mechanism of action without cross-resistance to clinical and marketed antimalarials.

The homopentameric PfFNT protein exhibits a protomer fold that mimics the aquaporin water and uncharged solute channels [6]. Despite its channel-like, rigid structure, PfFNT acts as a secondary-active transporter in the presence of a transmembrane proton gradient [2,7,8] by employing a unique “dielectric slide” transport mechanism in which the lactate substrate itself receives and carries the proton [9]. Human lactate/H^+^ transporters of the alternating-access monocarboxylate transporter family, MCT (SLC16A), are unrelated to PfFNT in terms of sequence, structure, and mechanism [10]. MMV007839 gave an IC_50_ of 170 nM on the PfFNT target when expressed and assayed for uptake of radiolabeled lactate in *Saccharomyces cerevisiae* yeast, and an EC_50_ of 140 nM with *Plasmodium falciparum* 3D7 parasites in vitro (IC_50_ 170 nM, EC_50_ 1.7 µM for MMV000972) [4]. MMV007839 has two tautomeric forms, i.e., a neutral hemiketal and a vinylogous acid that deprotonates to the anion [4]. The vinylogous acid tautomer represents the active inhibitor form, as shown by a compound, BH296 (Figure 1), that is incapable of forming the cyclic hemiketal due to the lack of a phenolic hydroxyl moiety yet was equally effective as MMV007839 [4]. The linear fluoroalkyl/vinylogous acid moiety of BH296 resembles two consecutive lactate molecules, indicating that the inhibitors act as substrate-analogs that bind deep in the PfFNT transport path.

Sub-lethal doses of MMV007839 selected resistant parasites that carried a single PfFNT G107S point mutation shifting the MMV007839 efficacy by two orders of magnitude (IC_50_ 21 µM, EC_50_ 35 µM) [4]. The mutation site is located towards the cytoplasmic entrance of the transporter, indicating that the compounds bind from the intracellular side to PfFNT. The inhibitor-bound protein structure was recently solved (PDB# 6vqr; 2.8 Å resolution) [11] and confirmed our binding model. We circumvented the PfFNT G107S resistance by a strictly linear vinylogous acid-type compound with a nitrogen atom in the aromatic ring for hydrogen bonding with the serine-hydroxyl, i.e., BH267.meta (Figure 1) [12]. The compound further blocks FNT-type lactate transporters of the remaining four human-pathogenic plasmodia at sub-micromolar concentrations [13].

The currently used yeast assay system is unsuitable for establishing affinity and kinetic binding parameters of the inhibitor compounds to PfFNT. This is due to the indirect nature of the transport inhibition determinations and the transmembrane diffusion component for the compounds to reach the cytoplasmic transporter binding site. Here, we set out to measure true binding affinities of the compounds for solubilized PfFNT wildtype and the G107S resistance mutation by fluorescence cross-correlation spectroscopy (FCCS) [14,15]. The method proved to be valid to kinetically characterize the interaction of the PfFNT inhibitors with its target (k_on_, k_off_, K_i_). While we found that the biophysical affinity data generally correlated well with the IC_50_ values obtained from functional assays in yeast, the FCCS approach revealed exceptions in which a compound exhibited high binding affinity yet showed low inhibitory efficacy in the yeast, most likely due to limited transmembrane diffusion.

## 2. Results

### 2.1. The Small Molecule Inhibitor BH296 Binds to the Cytoplasmic Side of PfFNT

We generated a HEK cell line that stably expresses PfFNT with a C-terminal GFP fusion. Placing GFP at the C-terminus was necessary to retain the membrane localization signal, mediating translocation into the ER and targeting to the plasma membrane. Confocal laser scanning microscopy indicated efficient targeting and incorporation of PfFNT-GFP into the plasma membrane of HEK cells (Figure 2). We functionalized BH296 with a 3-aminopropoxy moiety, replacing methoxy to facilitate labeling with the red fluorescent dye DY647 using succimidyl ester chemistry to generate a probe for imaging and spectroscopy (Figure 1).

BH296-DY647 was added at a concentration of 50 nM to cultured HEK cells stably expressing PfFNT-GFP to test for co-localization of the compound with the target at the plasma membrane by confocal imaging. However, even after incubation for 1 h, binding of labeled BH296 could not be observed (Figure 2a). As the cytoplasmic domain of the transporter is inaccessible for the compound, we next explored possibilities to permeabilize the plasma membrane and to retain cell viability for the time of incubation. For this, we used Triton X-100 at a final concentration of 0.05%, as described previously [16] to allow for diffusion of compounds into the cell. In the presence of Triton X-100, membrane localization of BH296-DY647 was clearly visible after 0.5 min incubation, and the intensity of staining increased within 5 min (Figure 2b,c). A prolonged incubation was impeded by incipient cell disintegration and membrane blebbing. We next set out to test the specificity of binding by analyzing the interaction of labeled BH296 with its unlabeled parental compound. As shown in Figure 2d, preincubation of cells with an excess of 100 µM of BH296 completely abrogated binding of the labeled inhibitor.

The monocarboxylate transporter hsMCT4 is a human functional homolog of PfFNT, but both proteins show no similarity on the basis of their primary sequence or transport mechanism. PfFNT, hsMCT4 was expressed as GFP fusion and expressed in HEK cells, but coincubation with BH296-DY647 did not show any co-localization on the plasma membrane (Appendix A). Taken together, these results show the specific and reversible binding of BH296 to the cytoplasmic domain of PfFNT in living cells.

### 2.2. BH267.meta, but Not BH296, Bind to the PfFNT Mutant G107S

A PfFNT G107S mutant generated by sublethal dosing of BH296 in vitro parasite cultures has gained attention as it confers resistance against BH296, but the introduction of a scaffold nitrogen atom (BH267.meta; Figure 1) restored the effectiveness of the inhibitor [11]. The G107S mutation site is located at the cytoplasmic entrance of PfFNT and narrows the transport path precluding the binding of BH296 via sterical hinderance of the phenyl moiety of the compound. Exchange of the carbon atom in the *meta*-position of the aromatic ring of BH296 for a nitrogen atom introduced a hydrogen bond acceptor site for the serine hydroxyl of the resistance mutation and restored binding to the mutant PfFNT G107S. We analyzed the interaction of BH267.meta by cellular imaging. For this, we generated labeled BH267.meta-DY647 (Figure 1), expressed PfFNT G107S in HEK cells, and incubated the cells with BH296-DY647 or BH 267.meta-DY647 after treatment with 0.05% Triton X-100. Comparison of cells stably expressing wildtype or mutant PfFNT-GFP showed similar expression levels and localization (Figure 3a,e).

As expected, at 50 nM, BH296 did not localize to membranes with PfFNT G107S, which is in line with the inactivity of the inhibitor against PfFNT G107S (Figure 3a). In contrast, BH267.meta-DY647 exhibited a clear membrane stain under identical conditions (Figure 3b). A direct comparison, however, revealed a lower intensity of staining than observed with BH296-DY647 on cells expressing PfFNT wildtype. The interaction could be competed with unlabeled BH267.meta but not with BH296 (Figure 3c,d). As suggested by the structure model [11], BH267.meta also binds efficiently to PfFNT wildtype, and the observed signal intensity of labeled BH267.meta-DY647 appears increased in direct comparison to mutant PfFNT. Whether these differences in signal intensity can be attributed to different affinities, to expression levels of the target protein, or to the cellular localization cannot be concluded from imaging-based experiments but requires biophysical analytical methods. However, the observation confirms previous findings that show the activity of BH267.meta on PfFNT G107S.

### 2.3. Fluorescence Cross-Correlation Spectroscopy (FCCS) Allows for Affinity Determinations of Drug Target Interactions

Still today, many approaches to characterize the interaction of inhibitors to their targets rely on purified proteins and interacting components, but many proteins exhibit their full activity only in the presence of binding partners and cofactors. Thus, characterizing interactions quantitatively and selectively in the full complexity of a cellular environment adds to the predictive value of the experiment yet is technically challenging. Fluorescence correlation spectroscopy (FCS) is a single molecule sensitive approach that exploits fluorescence fluctuations induced by low numbers of diffusing labeled particles in a confocal setup to analyze their concentrations and mobilities. Its dual color variant, FCCS, monitors molecular interactions by simultaneously analyzing diffusion events of differently labeled molecules through a microscopic open detection volume illuminated by overlapping lasers [14]. The approach allows for accurate measurements of affinities and rate constants of molecular interactions in cells, cellular lysates and other fluids or solutions [15]. In this investigation, we took advantage of the generated HEK cell lines that stably express PfFNT-GFP or PfFNT G107S-GFP and of the BH296 and BH267.meta compounds carrying red fluorescent DY647.

To this end, we prepared and solubilized HEK cell membranes comprising PfFNT-GFP or its G107S mutant. For optimal solubilization, a selection of detergents was tested at concentrations between 0.25% and 1%, aiming at conditions providing maximal solubilization efficiency and minimizing denaturing effects on the target protein. Moreover, the concentrations were chosen to remain below the critical micellar concentration, since detergent micelles exhibit the tendency to incorporate hydrophobic small molecule compounds and, thus, render them unavailable for interaction with the target protein. In the course of this investigation LMNG (1%), DDM (1%, 0.5%, 0.25%), DM (1%), CHAPS (1%) and LDAO (1%, 0.5%) were tested for membrane solubilization. Additionally, a compound mix of CHAPS (1%), DDM (0.5%) and CHS (0.2%) was tested. Testing these conditions identified LMNG at 1% to yield optimal performance with maximal complex formation of PfFNT with the fluorescent inhibitor probe as measured by FCCS and was used for the following experiments.

A total of 20 µL of solubilized membranes from HEK cells expressing PfFNT wildtype or G107S were transferred to a glass-bottom 384-well microtiter plate and analyzed for expression levels via GFP fluorescence intensity as well as diffusion times. This yielded a PfFNT protein concentration of 10 nM after membrane preparation and solubilization. The average dwell time was determined to 373 µs, which is in good agreement with membrane proteins of equal size. The signal fluctuations could be fitted to a 1-component diffusion model, indicating homogeneously monodisperse proteins (Appendix A).

Next, we determined inhibitor binding affinities for both PfFNT and its G107S mutant. Solubilized cell membranes containing PfFNT-GFP at a concentration of 10 nM were incubated for 1 h at ambient temperature with increasing concentrations of labeled inhibitor molecules to obtain saturation binding curves and to calculate the dissociation constant (K_D_) of the interactions. Equilibrium binding experiments were pipetted in biological triplicates and carried out and analyzed as described in Antoine et al. [15]. Data from the equilibrium saturation binding assays are shown in Figure 4, and calculated K_D_ values are summarized in Table 1. BH296 and BH267.meta bound to PfFNT wildtype with similar affinity of 67 and 72 nM (Figure 4a), respectively.

Irrespective of similar affinities that were calculated from the saturation binding curves, the obtained concentration of complexes differed substantially, with 6 and 3 nM for BH267.meta and BH296, respectively. The difference in complex concentration relates to a significantly reduced amount of binding competent PfFNT-GFP protein for BH296. Since the same sample of solubilized membranes was used for both tracer molecules, the result was counterintuitive, and we repeated the assay with varying concentrations of solubilized PfFNT-GFP. To exclude tracer instability, the batch used for saturation binding assays was re-purified on reversed phase HPLC; however, experiments repeated with quality-controlled tracers yielded identical results. In summary, the affinities calculated from the slopes of the recorded saturation binding curves are similar, but only a fraction of about half of PfFNT-GFP appears to display binding competence for the labeled variant of BH296. The smaller variant BH267.meta-DY647, in contrast, bound with a similar affinity to 100% of the available target (Figure 4a).

We next performed equilibrium saturation binding assays to assess the binding of both tracers to the mutant PfFNT G107S. Our results show weaker affinity by BH267.meta (K_D_: 405 nM), whereas BH296 failed to bind within detection limits (Figure 4b, Table 1), thereby reconfirming the observations of confocal imaging.

To analyze the binding properties of unlabeled inhibitors BH296 and BH 276.meta, we determined their inhibition constants (K_i_) in equilibrium competition binding assays. For this, we first incubated membrane preparations comprising 10 nM of PfFNT wildtype with labeled BH296 at a concentration of 70 nM. The interaction was then competed with increasing amounts of unlabeled parental compounds BH296 and BH267.meta within a concentration range of 10 pM and 10 µM. The concentration of formed complex between PfFNT-GFP and BH296-DY647 relates to the values observed in the saturation binding assays. As shown in Figure 4c, the competition curves exhibit similar IC_50_ values for both inhibitors. The corresponding K_i_ values were then calculated employing the Cheng Prusoff equation [17]. Table 2 shows the K_i_ values for both unlabeled inhibitors to the PfFNT wildtype, indicating similar high affinity to its target. The small difference in affinity compared to the results of saturation binding assays indicates that despite its bulkiness, the fluorophore does not impact the interaction to the target significantly.

In view of the differences in complex formation of both tracers, we were also interested to see how both unlabeled compounds perform in competition experiments with BH267.meta-DY647 as a probe. The identical PfFNT-GFP membrane preparation as in the previous experiment was incubated with BH267.meta-DY647 at a concentration of 70 nM to allow for complex formation. Different amounts of competitors BH296 and BH267.meta were added to the interaction at concentrations ranging from 10 pM to 10 µM and incubated at room temperature for 1 h to allow the interaction to reach equilibrium. BH267 meta replaced the labeled tracer with a K_i_ of 56 nM, and despite a residual background signal, the tracer was completely dissociated from the target. In contrast, BH296 was unable to entirely replace the labeled BH267.meta tracer molecule, despite its comparable observed K_i_ of 47 nM. A significant amount of 20% of complex remained intact, independent of competitor concentration and incubation time. This finding is of interest because it mirrors the behavior of BH296-DY647 in saturation binding assays, where a subfraction of target appeared incapable of binding to the more space-occupying BH296 (Figure 4d, Table 2).

The K_i_ values calculated for the interaction of BH296 and BH267.meta to mutant PfFNT G107S confirm the affinities observed for the direct interaction with labeled inhibitors. BH296 does not interfere with the binding of labeled BH267.meta to PfFNT G107S, but parental BH267.meta competes in this interaction with a K_i_ of 417 nM (Figure 4e, Table 2).

Together, the findings on inhibitor affinity correlate with the previous microscopic observations, which indicated high signal intensities for labeled BH296 and BH267.meta binding to PfFNT wildtype, no binding of BH296 to PfFNT G107S, while BH267.meta-DY647 exhibited binding to PfFNT G107S.

### 2.4. Binding Kinetics of Labeled BH296 and 267.meta to PfFNT by Time-Resolved FCCS Measurements

To complete our biophysical interaction analysis, we determined the rate constants of both labeled inhibitors to PfFNT wildtype. This is of relevance because inhibitors with similar affinities but different association and dissociation rates can show different activity profiles. The time a compound resides on its target, i.e., the drug target residence time, is defined as the reciprocal value of the dissociation rate constant (1/k_off_) and has been shown to be of considerable importance for target selectivity and duration of its inhibitory effect. For measuring the kinetics of direct interactions between PfFNT and the labeled inhibitor molecules, we applied the target at a concentration of 10 nM and chose ligand concentrations that exceeded the number of target molecules by a factor of >10. In this way, we ensured that throughout the course of the kinetic measurement, the depletion of free ligands remained insignificant. First, time resolved binding curves were recorded for ligand concentrations of 148, 133, 115, 100, 81, 76, 44, 25 nM (BH296) and 120, 102, 76, 74, 65, 58, 46, 32 nM (BH267.meta), respectively, to obtain k_obs_ values (Figure 5). Then k_on_ and k_off_ values were calculated by linear regression. Both inhibitors bind and dissociate with equal rate constants (Table 3) and are characterized by a rapid association. Equilibrium is reached within less than 1 min at nanomolar inhibitor concentrations. To cross-validate the reliability of our kinetic determinations, we compared the K_D_ values derived from saturation binding assays with affinities deduced from the kinetic rate constants by dividing k_off_ by k_on_. The obtained K_D_ of 113 nM for BH296 correlates with robust accuracy to the K_D_ previously measured by saturation binding 67 nM. Similarly, the calculated affinities for BH267.meta are in good agreement as well (112 and 72 nM, respectively; Table 3).

### 2.5. A comparison of Biophysical Affinity Data and IC_50_ Values from Functional Assays

With a loss of signal screening assay using the GFP fusion of PfFNT wildtype and BH296-DY647 as the interaction pair, we set out to test the FCCS approach on a selection of 26 compounds that were validated before using functional lactate transport assays in yeast [3,11]. In more detail, HEK cell lysates were adjusted to a PfFNT-GFP concentration of 4 nM and mixed with fluorescence-labeled BH296-DY647 at a final concentration of 80 nM, approximating the previously determined K_D_-value of this interaction. A total of 20 µL of the mix was then dispensed into the wells of a glass-bottom 384 microtiter plate. Subsequently, a dilution series covering a concentration range of 10 pM to 10 µM of each test compound was added and incubated for 1 h at room temperature for the interactions to reach equilibrium. The wells were then analyzed by FCCS, K_i_ values were determined, and compared by a scatter plot to the IC_50_ values from the previous functional assays in yeast (Figure 6). While the absolute numbers differed somewhat between the two independent assay types, we found a good correlation with respect to the compound ranking.

Solely BH454 posed an exception by showing little activity in the functional yeast assay despite an observed high affinity to PfFNT in the double-digit nanomolar range. This finding may be due to low transmembrane diffusion of the rather large compound in the yeast system. The deduced coefficient (r^2^ = 0.6) confirms the correlation of the data from the functional yeast assay and the direct interaction measurements, while the differences in the absolute IC_50_ and K_i_ values reflect a less-well defined assay format of a cellular assay, which lacks knowledge on the concentration of active PfFNT, and requires the inhibitor compounds to pass the plasma membrane either by diffusion or via transport proteins.

## 3. Discussion

The G107S point mutation arising after sublethal dosing of MMV007839 confers resistance against the drug and shifts the observed EC_50_ values by several orders of magnitude, as shown in functional assays. The exchange from a glycine to a serine residue narrows the diameter at the cytoplasmic entrance of the transporter. As such, we were interested to see whether the decrease in the functional activity of BH296 indeed relates to altered binding affinity to mutant PfFNT and precludes binding. Incubation of HEK cells stably expressing wild type PfFNT-GFP with a fluorescent derivative BH296-DY647 did not result in membrane localization of the compound at first, but permeabilization with low a concentration of Triton X-100 facilitated the interaction with the transporter. The interaction was shown to be reversible as competition with unlabeled inhibitors displaced the tracer. This finding reconfirms the cytoplasmic binding site for the inhibitor to PfFNT in living cells. Notably, incubation of fluorescent BH296 with HEK cells expressing mutant PfFNT G107S irrespective of membrane permeabilization failed to induce membrane localization. Treatment with detergents is detrimental to cellular viability and cells start to disintegrate after 5 min. Thus, incubation with fluorescent tracers is feasible only for a short time frame and interactions of inhibitors with substantially slower rates would be difficult to show by this imaging approach. However, BH267.meta-DY647, which occupies less space within the binding pocket, clearly binds to mutant PfFNT-GFP, albeit with weaker affinity. Taken together, our findings support the proposed binding mode of both inhibitors, BH296 and BH267.meta, and the inactivity of BH296 can be attributed to the deficiency of binding to mutant PfFNT.

To gain direct biophysical insight into the affinity and binding kinetics of small-molecule inhibitors to wildtype and mutant PfFNT, we employed FCCS as a drug discovery tool. Understanding the molecular interactions of an inhibitor to its designated target is critical to all stages of the drug discovery and development process. In functional cellular assays, such as the previously used yeast-based monitoring of PfFNT lactate transport, the activity readout of a compound is not purely defined by its target binding strength but by the additional contribution of indirect factors, such as solubility and membrane permeability. Here, our biophysical FCCS approach allowed for an unbiased look at the affinity and the binding kinetics. As such, biophysical measurements are important to guide compound optimization based on structure–activity relationships. The affinity of a drug to its target can be assessed only with knowledge of the active target concentration, yet this information is usually absent in cell-based functional assays. Here, FCCS yields comprehensive information on the concentrations of the target and tracer, homogeneity of particle size or the existence of aggregates, diffusion time and binding state, allowing for an accurate characterization of the molecular interaction. Moreover, precise determinations of rate constants help to select compounds with a long residence time for improved efficacy of lactate export inhibition. As such, functional testing, image-based co-localization studies, and FCCS-based biophysical characterization of the binding properties yield a comprehensive understanding of the behavior of the inhibitor in its physiological habitat. The short assay development time and the applicability of FCCS as a high throughput tool also facilitates the screening of a large number of compounds for their ability to interact with PfFNT wildtype and therapeutically relevant mutants.

The scatter plot comparing IC_50_ values from a yeast lactate uptake assay with FCCS derived K_i_ values shows good overall correlation but with a higher FCCS-derived affinity as the IC_50_ values would suggest. One compound, BH454, scored with an affinity below 100 nM in the FCCS assay but was not very effective in blocking lactate transport in the yeast system. To demonstrate activity in the yeast lactate transport assay, an inhibitor molecule has to pass the cell wall and cellular membrane in order to access and block the cytoplasmic target binding side. BH454 is a structurally symmetric compound, which carries two vinylogous acid moieties (Figure 6) that can both enter the binding pocket on PfFNT. With this, the compound has two binding sites, both of which mirror the BH454 structure, and similar binding affinity and inhibitory potency on lactate transport would have been expected. The difference in the observed biophysical affinity is a result of a decreased cellular uptake of the more polar compound in the yeast assay format.

## 4. Materials and Methods

### 4.1. Synthesis of BH296 and BH267.meta with a 3-Aminopropoxy Linker for Fluorescence Labeling

BH296 (4,4,5,5,5-pentafluoro-3-hydroxy-1-(4-hydroxyphenyl)pent-2-en-1-one) and BH267.meta carrying a hydroxyl moiety at the pyridine (4,4,5,5,5-pentafluoro-3-hydroxy-1-(6-hydroxypyridin-3-yl)pent-2-en-1-one) for attachment of the 3-aminopropoxy linker were synthesized as described before by a Claisen-type condensation in anhydrous THF using lithium hydride as a base [11]. Generally, for structure analysis and purity assessment (>95% of all compounds), mass spectrometry (LC-MS; Bruker Amazon SL) and nuclear magnetic resonance (Bruker Avance III 300) were employed.

A total of 1.13 g (4 mmol) of BH296 and 2.61 g (8 mmol) of cesium carbonate were suspended in 10 mL of *N,N*-dimethylformamide, stirred for 30 min at room temperature upon addition of 3-(Boc-amino)propyl bromide dissolved in 5 mL of *N,N*-dimethylformamide. The reaction was kept stirring for 16 h, before it was filtered, 1 mL of concentrated acidic acid was added to the filtrate, and the solvent was evaporated off. Silica gel chromatography with cyclohexane/ethyl acetate (8:2) as the mobile phase yielded 77% of the desired product (C_19_H_22_F_5_NO_5_; 439.38 g mol^−1^).

^1^H-NMR (300 MHz, 25 °C, [*d*_6_]-DMSO): 

δ/ppm = 1.38 (s, 9H, O-C(CH_3_)_3_); 1.83–1.90 (m, 2H, -CH_2_); 3.07–3.12 (m, 2H, -CH_2_); 4.10–4.13 (*t*, ^3^*J* = 6.3 Hz, 2H, -CH_2_); 6.92–6.95 (m, 1H); 7.00 (*s*, 1H); 7.10 (*d*, ^3^*J* = 9.0 Hz, 2H); 8.16 (*d*, ^3^*J* = 9.0 Hz, 2H).

The protective Boc group was removed by dissolving 1.36 g of product in 4 mL of dichloromethane, dropwise addition of 4 mL of trifluoro acetic acid, and stirring at room temperature for 5 h. The solvent was evaporated off, yielding 95% of the product (C_18_H_21_F_5_N_2_O_5_; 440.37g mol^−1^).

^1^H-NMR (300 MHz, 25 °C, [*d*_6_]-DMSO): 

*δ*/ppm = 1.37 (*s*, 9H, O-C(CH_3_)_3_); 1.75–1.81 (*m*, 2H, -CH_2_); 2.94–2.99 (*m*, 2H, -CH_2_); 3.96–3.99 (*t*, ^3^*J* = 7.0 Hz, 2H, -CH_2_); 6.52 (*d*, ^3^*J* = 9.8 Hz 1H); 6.93 (*t*, ^3^*J* = 6.9 Hz 1H); 6.98 (*s*, 1H); 7.99 (*dd*, 1H); 8.98 (*d*, ^3^*J* = 2.5 Hz 1H).

BH267.meta carrying a 3-aminopropoxy linker was synthesized using the same procedure, starting from 0.3 g of 4,4,5,5,5-pentafluoro-3-hydroxy-1-(6-hydroxypyridin-3-yl)pent-2-en-1-one to yield 64% of the Boc-protected product (C_18_H_21_F_5_N_2_O_5_; 440.37g mol^−1^).

^1^H-NMR (300 MHz, 25 °C, [*d*_6_]-DMSO):

*δ*/ppm = 1.37 (*s*, 9H, O-C(CH_3_)_3_); 1.75–1.81 (*m*, 2H, -CH_2_); 2.94–2.99 (*m*, 2H, -CH_2_); 3.96-3.99 (*t*, ^3^*J* = 7.0 Hz, 2H, -CH_2_); 6.52 (*d*, ^3^*J* = 9.8 Hz 1H); 6.93 (*t*, ^3^*J* = 6.9 Hz 1H); 6.98 (*s*, 1H); 7.99 (*dd*, 1H); 8.98 (*d*, ^3^*J* = 2.5 Hz 1H).

Removal of the Boc group by trifluoro acetic acid treatment yielded 87% of BH267.meta carrying a 3-aminopropoxy linker (C_16_H_15_F_8_NO_5_; 454.27 g mol^−1^).

^1^H-NMR (300 MHz, 25 °C, [*d*_6_]-DMSO):

*δ*/ppm = 1.95–2.03 (*m*, 2H, -CH_2_); 2.78–2.86 (*m*, 2H, -CH_2_); 4.07 (*t*, ^3^*J* = 6.9 Hz, 2H, -CH_2_); 6.55 (*d*, ^3^*J* = 9.7 Hz 1H); 6.95 (*s*, 1H); 7.84 (*s*,3H, -NH_3_); 8.03 (*dd*, 1H); 8.98 (*d*, ^3^*J* = 2.5 Hz 1H).

### 4.2. Cell Culture Conditions

The human embryonic kidney HEK293 cells were cultured in Dulbecco’s Modified Eagle Medium (DMEM) with high glucose (cat. no 41966-029; Gibco, UK) supplemented with 10% (*v*/*v*) fetal calf serum (cat. no S0115; Biochrom, Germany), 100 units/mL of penicillin and 100 µg/mL of streptomycin (cat. no 15140-122; Gibco, UK). The cell line was stably transfected with a TET-repressor encoding sequence, from the pcDNA™6/TR vector (cat. no V102520; Invitrogen, USA), for tetracycline-regulated expression of the gene of interest. The TET repressor was maintained in the cells by growing them in media containing 5 µg/mL of blasticidin (cat. no A3784.0025; Applichem, Germany). The cells were grown to 80% confluency per cell culture dish (cat. no 83.3902; Sarsted, Germany) at 37 °C and 5% CO_2_ in a humidified incubator.

### 4.3. Expression of PfFNT-GFP

The DNA of full-length PfFNT from *Plasmodium falciparum* 3D7 was cloned as a C-terminal GFP fusion in mammalian expression vector pCGTO (a derivative of pcDNA3.1, cat. no V86020, Invitrogen, USA). The point mutation (G107S) was introduced into PfFNT using the Q5^®^ Site-Directed Mutagenesis Kit (cat. no E0554S; NEB, USA). All constructs were confirmed by DNA sequencing. The PfFNT WT and G107S proteins were expressed in HEK293 cells, under the control of the TET promoter, using a stable expression system. HEK293 cells (5 × 10^6^) were transfected with 8 μg of plasmids encoding the GFP-tagged protein using the jetPRIME^®^ Transfection Kit (cat. no 114-75; Polyplus-transfection^®^, France). To select the stable transfected cells, the selection antibiotic zeocin (cat. no R250-01, LifeTechnologies, USA) was used at a concentration of 100 µg/mL. Around 14 h before imaging or harvesting the cells, PfFNT expression was induced by adding 1 µg/mL of tetracycline to the media. During the harvest, the cells were washed twice with PBS, pelleted for 5 min at 1100 g, and then frozen at −80 °C.

### 4.4. Generation of Cell Lysate

A cell pellet was resuspended in 1 mL of ice-cold TBS (50 mM Tris-HCl (pH 7.4) and 150 mM NaCl) supplemented with cOmplete™ Protease Inhibitor Cocktail (cat. no 11697498001, Roche, Switzerland) and broken by sonication on ice, using an Ultrasonic homogenizer SONOPULS HD 2070 (Bandelin, Germany). Cell debris was removed by centrifugation for 5 min at 1100× *g* at 4 °C, and membranes were harvested from the cleared lysate by subsequent centrifugation at 21,000× *g*. The resulting pellet containing cellular membranes was lysed using a Dounce homogenizer. The receptor was solubilized by incubation in 1% LMNG (cat. no NG310, Anatrace, USA) or in another detergent (DDM, DM, CHAPS, CHS, LDAO) when conducting tests to determine the best solubilization conditions. The insolubilized membrane material was removed by centrifugation for 1 h at 21,000× *g* at 4 °C, and the supernatant was collected and directly subjected to FCCS analysis or stored at −80 °C.

### 4.5. Fluorescent Labeling of the Tracer Molecule

The compounds used for fluorescent labeling were resuspended in DMSO to a final concentration of 50 nM. Then, 4 µL (200 nmol) of these compounds were dissolved in 95.3 µL of DMSO + 0.7µL of DIPEA and labeled via the amino group to DY647-Peg4 (0.2 mg, 200 nmol) (cat. no 647P1-01, Dyomics, Germany) utilizing a reactive NHS ester group. The mixture was incubated in the dark at room temperature (RT) for 2 h. The conjugate was purified on a reversed-phase high-performance liquid chromatography (HP-1100, Agilent, USA) using an ACN/H_2_O gradient from 20% to 80% ACN (BH296) or 40% to 60% ACN (BH267.meta), allowing separation of unlabeled and labeled compounds. Then, the labeled compound was lyophilized, dissolved in DMSO, and stored at −20 °C.

### 4.6. Live Cell Imaging

The confocal laser scanning microscopy was performed on an LSM 510 confocal microscope connected to an Axiovert 200M equipped with a C-Apochromat 40×/1.2 W water immersion objective (Carl Zeiss, Germany). The red fluorophore (Dy-647) was excited using a 633 nm helium–neon laser, whereas the green fluorophore (GFP) was excited using a 488 nm argon-ion laser.

### 4.7. FCCS Analysis

FCCS measurements were conducted in 384-well glass-bottom plates (cat. no PS96B-G175, SWISSCI, Switzerland) using an Insight plate reader (Evotec Technologies, Germany) fitted with a U-Apo300 40x water immersion lens, NA 1.15 (Olympus, Japan). The 488 nm laser line of an argon-ion laser and the light of a 633 nm helium–neon laser were used for excitation. Fluorescence fluctuations were recorded for 8 s with 12 repetitions and analyzed afterward using FCS+plus Analyze 1.1P (Evotec technologies, Version 1.2.5g). FCCS analysis was carried out as described previously [15]. Briefly, for measuring the dissociation constant (K_D_), a titration series of 12 dilutions was prepared, where the concentration of the lysates obtained from HEK cells expressing GFP-fusions of PfFNT WT and G107S was kept constant, and the concentration of the labeled binding partner (BH296-DY647 and BH267.meta-DY647, respectively) was varied. The IC_50_ values were calculated by plotting the concentration from the dual labeled complex against the concentration of the competitor titrated over a concentration range from 10 µM to 10 pM. Samples were incubated for 1 h at room temperature prior to FCCS measurements to allow the formation of drug target complexes. From the resulting IC_50_ values, the K_i_ values were obtained by application of the Cheng Prusoff equation [17]. To assess the rate constants, time-resolved measurements were carried out at different concentrations of the labeled ligand. By plotting different rate constants (k_obs_) against the concentration of the labeled ligand, the k_obs_ values resulted in a straight line, the slope of which is k_on_ value and the y-intercept the k_off_ value. All K_D_ and IC_50_ values, as well as kinetic parameters, were determined from three independently repeated experiments. The figures show one representative measurement; however, the calculated K_D_, K_i_, k_on_ and k_off_ values are presented as an average of the three measurements ± SD. Error bars denote ± SEM.

## 5. Conclusions

The use of the FCCS assay will guide future medicinal chemistry efforts to improve the binding affinities of inhibitors to PfFNT based solely on the real interaction with the target. Thus, using the established FCCS assay will help to further optimize PfFNT inhibitors for improved affinity and long residence times.

## Figures and Tables

**Figure 1 pharmaceuticals-14-00757-f001:**
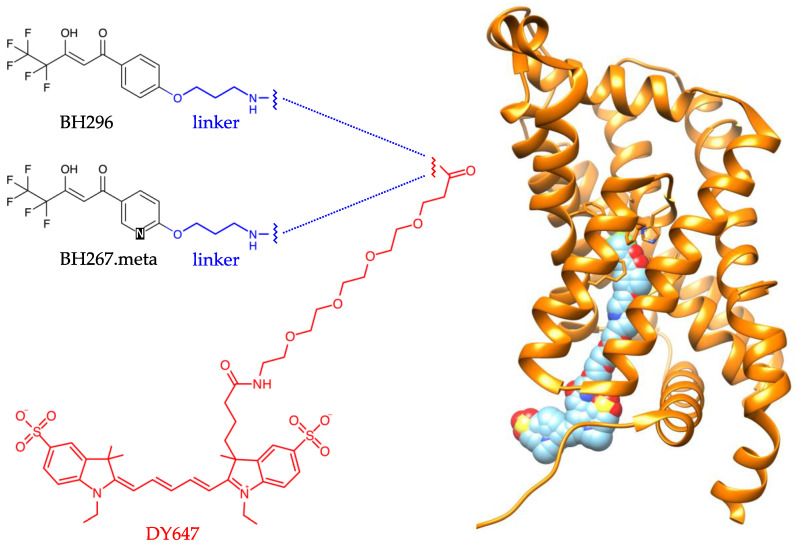
Inhibitors of PfFNT (BH296, BH267.meta) fused with a fluorescence label (DY647) via a 3-aminopropoxy linker. The 4× ethylene glycol unit of the label prevents collision of the fluorescent moiety with the protein target; see model of PfFNT (PDB# 6vqr) with bound inhibitor carrying the label (right panel).

**Figure 2 pharmaceuticals-14-00757-f002:**
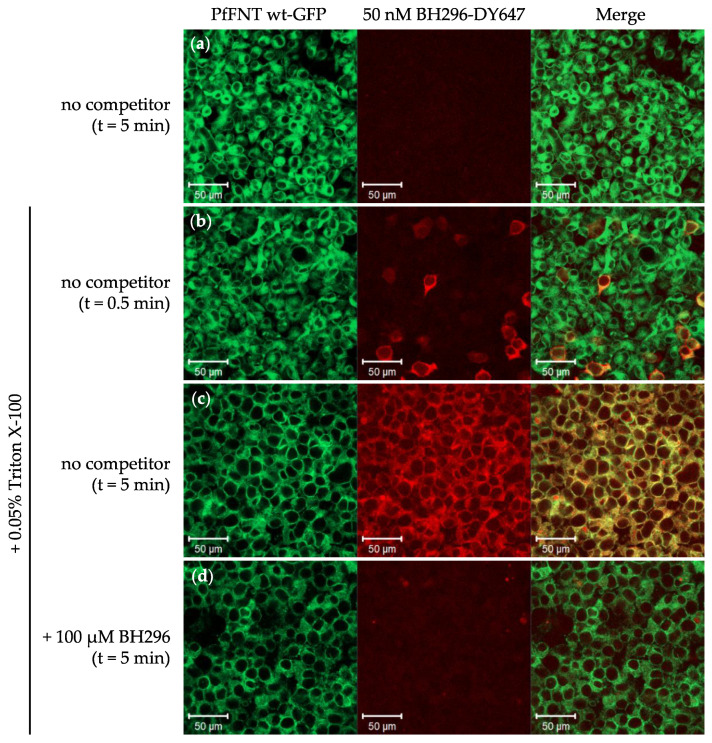
Live cell imaging of PfFNT-GFP stably expressed in HEK293 cells and binding of BH296-DY647. Representative images of PfFNT-GFP fusions (green, left panels) and BH296-DY647 (red, middle panels) are shown. (**a**) No co-localization of PfFNT-GFP and BH296-DY647 was observed in the absence of Triton X-100. Co-localization was clearly visible after treating the cells with 0.05% Triton X-100 for membrane permeabilization. After 30 s of incubation (**b**, merged images, co-localization in yellow, right panels), membrane localization of BH296-DY647 was clearly visible and increased over time (**c**, incubation time 5 min). (**d**) Preincubation with 100 µM of unlabeled BH296 prevented binding of the labeled tracer, indicating that co-localization is mediated by interaction with PfFNT-GFP.

**Figure 3 pharmaceuticals-14-00757-f003:**
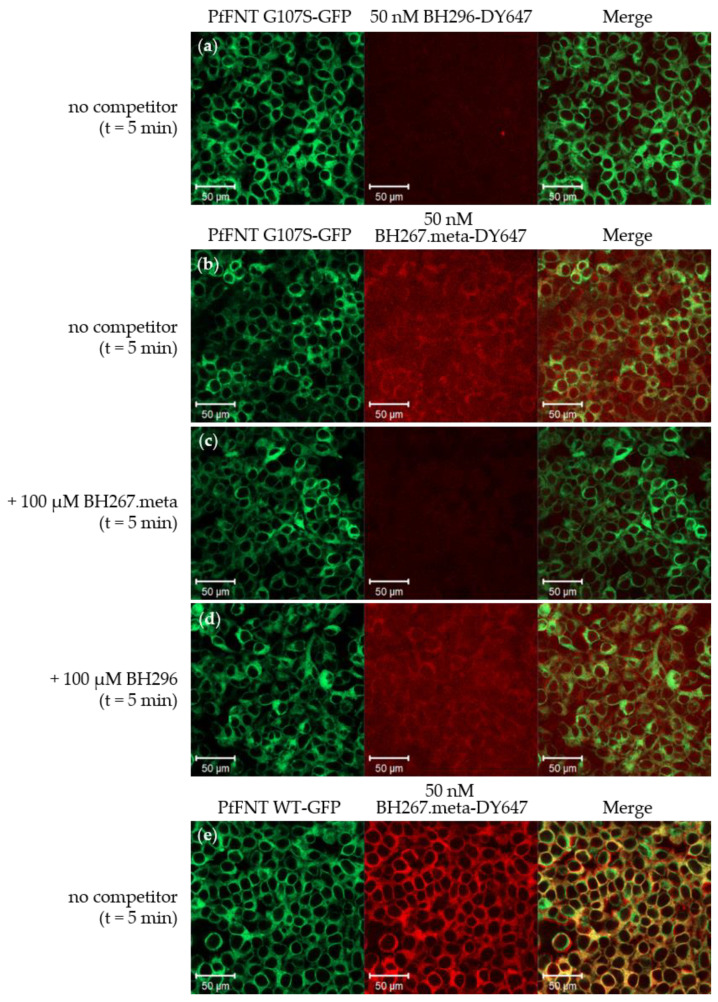
Live cell imaging of mutant PfFNT G107S-GFP stably expressed in HEK293 cells and binding of labeled ligands BH296-DY647 and BH267.meta-DY647. BH267.meta-DY647 (**b**), but not BH296-DY647 (**a**) co-localizes with the membrane-bound G107S mutant of PfFNT-GFP, and competition with 100 µM BH267.meta prevents membrane localization (**c**). Based on fluorescence intensity levels, the interaction appears to be weaker than for the wild type PfFNT-GFP to BH296-DY647 (see Figure 1b,c). (**d**) The interaction cannot compete with BH296. (**e**) BH267.meta-DY647 localizes with PfFNT wildtype. Incomplete overlap of both channels in E does not represent altered localization but can be attributed to an offset between both channels and thus is irrelevant for the drawn conclusion of the experiment.

**Figure 4 pharmaceuticals-14-00757-f004:**
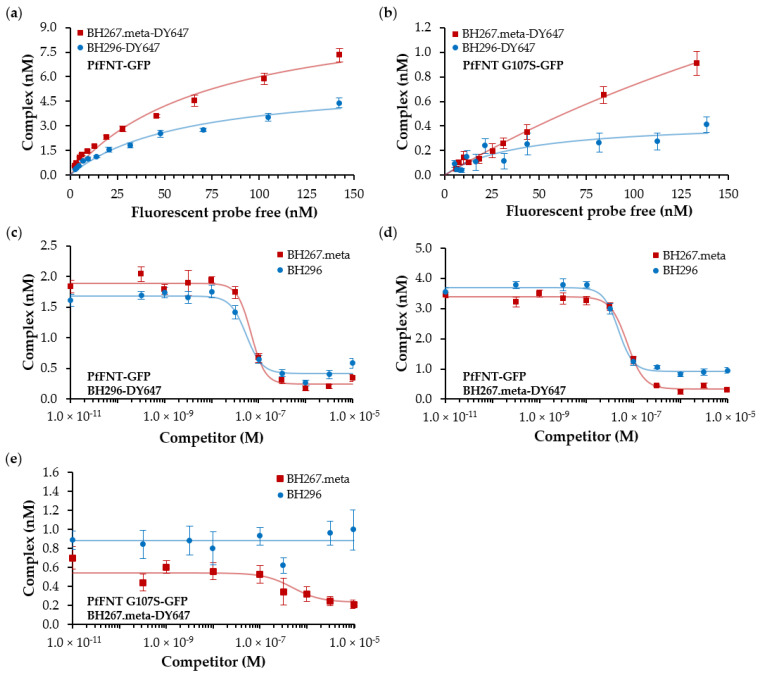
Affinity determination of BH296, BH267.meta to PfFNT and PfFNT G107S, and equilibrium saturation binding of BH296 and BH267.meta to PfFNT and PfFNT G107S. (**a**) BH296-DY647 and BH267.meta-DY647 bound with similar affinity to PfFNT (67 nM and 72 nM, respectively), but binding to BH296-DY647 was saturated with only 50% of the target involved in interactions. BH267.meta bound to 100% of the GFP-fusions of wildtype PfFNT. (**b**) Saturation binding assays of BH296-DY647 to mutant PfFNT-GFP indicated no affinity above background caused by fluorescence cross talk, but BH267.meta-DY647 bound with a K_D_ of 405 nM. (**c**) Membrane preparations containing 10 nM of PfFNT-GFP were incubated with 70 nM BH296-DY647, and associated complexes competed with free compounds BH267.meta and BH296 to assess their K_i_ values. In the absence of a competitor, 1.5 nM of complex was formed, and both compounds displaced the tracer completely with a K_i_ of 48 and 66 nM, respectively. (**d**) Using BH267.meta-DY647 as a tracer, both compounds displaced the total of 3 nM complexes with K_i_ values of 47 and 56 nM. (**e**) Membrane preparations containing GFP fusions of mutant PfFNT G107S were mixed with 150 nM of the labeled BH267.meta tracer and competed with BH267.meta and BH296. Only BH267.meta displaced the tracer with a K_i_ of 417 nM, whereas BH296, even at concentrations of 10 µM, did not compete in the interaction.

**Figure 5 pharmaceuticals-14-00757-f005:**
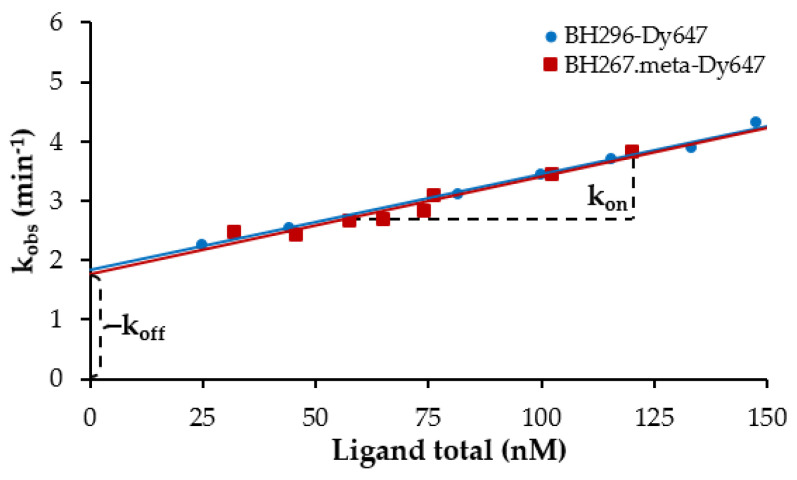
Kinetic studies of the fluorescent probes BH296-DY647 and BH267.meta-DY647 with PfFNT-GFP by FCCS. Time resolved measurements of the interactions were carried out to obtain k_obs_-values. The k_obs_-values were plotted versus concentrations of the fluorescent probe. The solid line represents the best linear regression, and k_on_ and k_off_ of both fluorescent probes were estimated from the slope and y-intercept, respectively. k_on_ and k_off_ values of BH296-DY647 and BH267.meta-DY647 are summarized in Table 3.

**Figure 6 pharmaceuticals-14-00757-f006:**
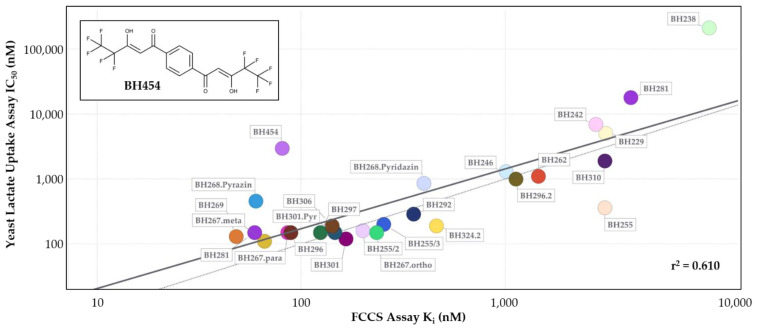
Scatter plot of FCCS-derived K_i_ values vs. IC_50_-values from the yeast lactate uptake assay. The activity of 26 compounds on the uptake of lactate in the yeast assay (y-axis) was plotted against the K_i_ values obtained from FCCS competition assays using the PfFNT-GFP and BH296-DY647 (x-axis). The diagonal is displayed as a thin dotted line, and regression is shown as a thick black line, r^2^: 0.610.

**Table 1 pharmaceuticals-14-00757-t001:** K_D_ values.

Target	Fluorescent Probe	K_D_ (nM)	Active Receptor (%)
PfFNT wt-GFP	BH296-DY647 BH267.meta-DY647	67 ± 6 72 ± 1	68 ± 3 100
PfFNT G107S-GFP	BH296-DY647 BH267.meta-DY647	>2000 405 ± 1	n.d. 100

**Table 2 pharmaceuticals-14-00757-t002:** K_i_ values.

Target	Fluorescent Probe	K_i_ Competitor (nM)	
		BH296	BH267.meta
PfFNT wt-GFP	BH296-DY647 BH267.meta-DY647	48 ± 2 47 ± 1	66 ± 6 56 ± 3
PfFNT G107S-GFP	BH267.meta-DY647	>10,000	417 ± 16

**Table 3 pharmaceuticals-14-00757-t003:** k_on_ and k_off_ values.

Target	Fluorescent Probe	k_on_ (nM^−1^ min^−1^)	k_off_ (min^−1^)	K_D_ (Kinetics, nM)	K_D_ (Equilibrium, nM)
PfFNT wt-GFP	BH296-DY647	0.0167 ± 0.0006	1.8659 ± 0.0339	113 ± 2	67 ± 6
	BH267.meta-DY647	0.0161 ± 0.0005	1.7839 ± 0.0250	112 ± 5	72 ± 1

## Data Availability

Data is contained within the article and Appendix A.

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
