# Peer review of "Fluorescence Cross-Correlation Spectroscopy Yields True Affinity and Binding Kinetics of Plasmodium Lactate Transport Inhibitors"

_pharmaceuticals, 2021, doi:10.3390/ph14080757_

Round 1

Reviewer 1 Report

This manuscript, "Fluorescence Cross-Correlation Spectroscopy...", by Jakobowska, et al. is a well-written and substantial manuscript. It is, in the opinion of this reviewer, worthy of publication essentially as-is.

A couple of minor points:

  1. 3 lines from the bottom of the caption of Figure 4 states that 'yy nM'  of the labeled BY267... This should be updated to be explicit.
  2. The 1H NMR results should be changed to 1H NMR throughout.

Author Response

  1. 3 lines from the bottom of the caption of Figure 4 states that 'yy nM' of the labeled BY267... This should be updated to be explicit.

We are very sorry for letting this pass in the final version. The correct concentration is 150 nM of the labelled BH267.

  1. The 1H NMR results should be changed to 1H NMR throughout.

Thank you for pointing it out. Changes have been made.

Reviewer 2 Report

I have no comments. I recommend it for publication at Pharmaceuticals.

Author Response

no comments were made by the reviewer

Reviewer 3 Report

The authors expressed PfFNT fused with a green fluorescent protein in human embryonic kidney cells and generated fluorescent derivatives of the inhibitors, BH296 and BH267.meta. By confocal imaging, they identified the binding location at the cytosolic transporter entry site; and carried out fluorescence cross correlation spectroscopy to study the binding constants. Finally, they found the robust correlation of the results from the biophysical FCCS binding assay to inhibitor of the functional transport assay. This work is interesting and guiding for the researchers in the correlated fields. I think it is can be accepted by this journal after minor alteration.

(1) The authors are supposed to provide high resolution images such as Figure 6; it is too much vague to get significant information for the readers.

(2) In the Line 418, the authors calculated the amount of the chemicals, it seems the authors made the wrong calculation of molecular weights of Cesium carbonate, please confirm that.

Author Response

  1. The authors are supposed to provide high resolution images such as Figure 6; it is too much vague to get significant information for the readers.

We apologise that the image was unreadable and have now added a higher resolution image.

  1. In the Line 418, the authors calculated the amount of the chemicals, it seems the authors made the wrong calculation of molecular weights of Cesium carbonate, please confirm that.

You are right, the calculation gives millimol instead of nanomol: 2.61 g of cesium carbonate equal 8 mmol (2.61 g / 325.82 g/mol = 0.008 mol)

The same is true for 1.13 g of BH296, which are 4 mmol and not nanomols.